# Antibiotic Resistance Trends in Recurrent Paediatric Urinary Tract Infections: A Five-Year Single-Centre Experience

**DOI:** 10.3390/children12111567

**Published:** 2025-11-18

**Authors:** Olivia-Oana Stanciu, Mircea Andriescu, Andreea Moga, Ruxandra Caragata, Laura Balanescu, Radu Balanescu

**Affiliations:** 1Department of Paediatric Surgery and Orthopaedics, “Carol Davila” University of Medicine and Pharmacy, 050474 Bucharest, Romania; olivia.stanciu@drd.umfcd.ro (O.-O.S.); andreea.moga@umfcd.ro (A.M.); ruxandra.caragata@umfcd.ro (R.C.); laura.balanescu@umfcd.ro (L.B.); radu.balanescu@umfcd.ro (R.B.); 2Paediatric Surgery Department, “Grigore Alexandrescu” Clinical Emergency Hospital for Children, 011743 Bucharest, Romania

**Keywords:** paediatric urinary tract infection, recurrent UTI, antimicrobial resistance, multidrug resistance, ESBL, antibiotic prophylaxis, urinary tract malformations

## Abstract

**Background**: Recurrent urinary tract infections (rUTIs) in children are increasingly complicated by antimicrobial resistance, leading to limited treatment options and challenging prophylactic management. Continuous local monitoring of resistance trends is essential for evidence-based stewardship. **Methods**: This retrospective study analysed recurrent paediatric UTI recorded between 2020 and 2024 at a tertiary hospital in Romania. Data were extracted using the ICD-10 code N39.0 and included demographic, clinical, and microbiological variables. Antimicrobial susceptibility testing followed CLSI standards. Associations between multidrug resistance (MDR) and clinical factors were assessed with χ^2^ tests and Cramer’s V, and predictors of MDR were evaluated by multivariable logistic regression. Temporal trends in resistance were examined using logistic regression with year as a continuous variable, and results were validated with the non-parametric Cochran–Armitage linear-by-linear χ^2^ trend test to strengthen analytical rigour. Proportions are presented with Wilson 95% confidence intervals (CIs). **Results**: A total of 134 children met inclusion criteria for rUTI, of whom 130 had complete demographic and microbiological data and were included in analyses. Each episode represented a distinct culture-confirmed infection occurring ≥30 days apart. MDR occurred in 48.5% of isolates (95% CI, 40.2–56.9) and ESBL in 20.9% (95% CI, 14.9–28.5). MDR was significantly associated with urinary tract malformations (*χ*^2^ = 5.78, *p* = 0.016) and continuous antibiotic prophylaxis (*χ*^2^ = 4.23, *p* = 0.040). Neither logistic nor Cochran–Armitage trend analyses demonstrated a significant temporal increase in MDR (OR per year = 0.94; 95% CI 0.75–1.17; *p* = 0.566; χ^2^ = 0.89; *p* = 0.346). **Conclusions**: MDR and ESBL rates among children with recurrent UTIs remain high but stable. The combined use of parametric and non-parametric trend analyses confirmed the absence of a significant upward trajectory, underscoring the need for ongoing surveillance and stewardship to maintain antibiotic effectiveness in paediatric care.

## 1. Introduction

Urinary tract infections (UTIs) are among the most common bacterial infections in childhood, affecting up to 8% of girls and 2% of boys during the first eight years of life [1]. Although most episodes resolve with appropriate antimicrobial therapy, a substantial proportion of children experience recurrent UTIs (rUTIs), which are defined as two or more febrile or three or more culture-confirmed infections within one year [2]. Recurrent infections are clinically significant because they increase the risk of renal scarring, hypertension, and long-term impairment of renal function, particularly in the presence of vesicoureteral reflux or other urological anomalies.

Effective management of rUTIs relies heavily on the use of empirical antibiotic therapy guided by local susceptibility patterns. However, the global escalation in antimicrobial resistance (AMR) has substantially reduced the effectiveness of commonly used agents such as amoxicillin–clavulanate and trimethoprim–sulfamethoxazole [3]. Multidrug-resistant *Escherichia coli*, *Klebsiella* spp., and other Gram-negative uropathogens are now being reported with increasing frequency even in community-acquired infections, complicating both prophylactic and therapeutic strategies in paediatric populations [4,5].

Children with rUTIs are particularly vulnerable to infection with resistant organisms due to repeated antibiotic exposure, prior hospitalizations, and underlying anatomical or functional urinary tract abnormalities [6]. Yet, most surveillance studies focus on first-time UTIs and data on resistance patterns specifically in recurrent paediatric infections remain limited—particularly in Central and Eastern Europe, where surveillance networks and antibiotic stewardship programmes are still developing [5,7].

To date, no Romanian or regional studies have longitudinally characterised antimicrobial resistance exclusively in children with recurrent UTIs. Prior regional reports described resistance in broader paediatric UTI cohorts or organism-specific analyses [6,8,9]. Existing reports either have combined first-episode and recurrent cases or focus exclusively on community-acquired infections, thereby overlooking the unique microbiological dynamics that develop in patients repeatedly exposed to antimicrobials. This lack of targeted data limits the ability to tailor empirical therapy and prophylactic strategies for high-risk paediatric populations in this region. Recurrent UTIs differ fundamentally from first infections in both pathophysiology and resistance selection. Repeated antibiotic exposure can alter the urinary microbiota, favour colonisation by multidrug-resistant (MDR) strains, and facilitate biofilm formation within the urinary tract. Moreover, structural or functional urinary anomalies often result in incomplete bacterial clearance, creating a persistent reservoir for resistant organisms. These factors make recurrent infections not only more difficult to treat but also an important source of antimicrobial resistance dissemination.

Based on these considerations, we hypothesised that children with urinary tract malformations and those receiving continuous antibiotic prophylaxis (CAP) would exhibit higher rates of multidrug-resistant (MDR) and extended-spectrum β-lactamase (ESBL)-producing uropathogens. We further anticipated that resistance prevalence would increase progressively over the study period, reflecting cumulative antibiotic selection pressure in this vulnerable cohort.

Extended-spectrum β-lactamase (ESBL)–producing Enterobacteriaceae have emerged as a major cause of paediatric urinary tract infections (UTIs) worldwide, posing significant therapeutic and epidemiological challenges. Reported ESBL prevalence in children varies between 10% and 35%, depending on geographic region, healthcare exposure, and prior antibiotic use [10,11,12]. These infections are associated with higher recurrence rates, prolonged hospitalisation, and an increased need for parenteral or carbapenem therapy due to limited oral treatment options [13]. In community settings, *Escherichia coli* and *Klebsiella* species remain the predominant ESBL producers, reflecting the spread of plasmid-mediated resistance even among previously healthy children [9,10]. Despite growing global awareness, data on paediatric ESBL prevalence in Central and Eastern Europe remain scarce, underscoring the importance of regional surveillance studies such as the present one to guide empirical therapy and antibiotic stewardship initiatives.

Recurrent UTIs in children most commonly reflect an underlying predisposing condition rather than random reinfection. Key structural causes include congenital anomalies of the kidney and urinary tract (CAKUT), particularly vesicoureteral reflux, obstructive uropathies (e.g., ureteropelvic junction obstruction, primary obstructive megaureter), duplex collecting systems with ureterocele, and posterior urethral valves, all of which impair urinary drainage and facilitate bacterial persistence. Functional abnormalities such as bladder–bowel dysfunction, dysfunctional voiding, neurogenic bladder, and chronic constipation further increase the risk of recurrence and breakthrough infections despite prophylaxis. In addition, suboptimal or prolonged antibiotic prophylaxis, prior broad-spectrum antibiotic exposure, poor perineal hygiene, and sexual activity in adolescents may contribute to recurrent episodes and select for multidrug-resistant uropathogens. Recent Romanian and regional data reporting high resistance rates among children with CAKUT support the central role of structural and functional abnormalities as drivers of both recurrence and antimicrobial resistance in paediatric UTIs [14,15,16].

Understanding local resistance trends over time is critical to guide empiric treatment, optimise prophylaxis, and support antimicrobial stewardship. This study provides the first longitudinal analysis of recurrent paediatric UTI resistance trends in Romania, offering local data to inform regional stewardship policies. Prior Romanian reports focusing on CAKUT-associated UTIs have identified comparable resistance patterns, particularly among *E. coli* and *Klebsiella* isolates [6,8,17], though none have evaluated temporal evolution or specifically addressed recurrent infections.

## 2. Materials and Methods

### 2.1. Study Design and Ethical Approval

This retrospective observational study was conducted at the Department of Paediatrics and Paediatric Surgery, “Grigore Alexandrescu” Emergency Hospital for Children, a tertiary referral centre in Bucharest, Romania. The study period spanned January 2020 to December 2024, and included paediatric patients aged 0–18 years diagnosed with recurrent urinary tract infections (rUTIs). The study adhered to the principles of the Declaration of Helsinki (revised 2013) and complied with the European Union General Data Protection Regulation (GDPR 2016/679) [18]. Ethical approval was granted by the Ethical Committee of “Grigore Alexandrescu” Emergency Hospital for Children. The requirement for individual informed consent was waived due to the retrospective nature of the study.

### 2.2. Study Population

Hospital electronic medical records were queried using the ICD-10 diagnostic code N39.0 (“urinary tract infection”), yielding 708 paediatric cases.

After applying predefined inclusion and exclusion criteria, 134 rUTI episodes from paediatric patients met eligibility requirements, defined as follows: two or more febrile UTIs [19] or three or more culture-confirmed UTIs within 12 months [19]. Recurrent infection episodes were included as independent events only if they occurred ≥30 days apart and yielded distinct culture-confirmed isolates [19]. This interval was chosen to distinguish new infection episodes from treatment failure or relapse, consistent with prior paediatric UTI studies [20].

Exclusion criteria included: mixed bacterial growth or contaminated samples, incomplete microbiological data, and catheter-associated or nosocomial UTIs. Incomplete microbiology data referred to episodes where the urine culture yielded growth but antibiotic susceptibility or species identification was unavailable due to laboratory processing errors or incomplete electronic entry. At our laboratory, samples with mixed growth of ≥3 species or colony counts below 10^5^ CFU/mL (in the setting of ≥3 species) are considered contaminated and excluded, based on local laboratory standard operating procedures and published literature suggesting that heavy mixed growth rarely represents true paediatric UTI [21].

Of these, 130 episodes (from 130 children) had complete demographic and microbiological data and were included in the final analyses.

### 2.3. Data Collection

For each episode, the following data were extracted from the hospital microbiology and clinical databases:Demographics: age (years), sex, anonymized identification number;Clinical factors: presence and type of urinary tract malformation, presence of other congenital anomalies, and use of continuous antibiotic prophylaxis at the time of infection;Microbiological data: bacterial species, extended-spectrum β-lactamase (ESBL) status, and antibiotic susceptibility results.

Patient age was calculated as the interval between the date of birth derived from the national personal identification number (CNP) and the date of urine culture collection (infection episode).

The presence and type of congenital anomalies of the kidney and urinary tract (CAKUT) were confirmed through available imaging studies (renal and bladder ultrasound, voiding cystourethrography, or dimercaptosuccinic acid [DMSA] scintigraphy) documented in the patient’s medical record. In most cases, CAKUT had been previously diagnosed during earlier urological evaluations; no additional imaging was performed solely for this study.

All data were de-identified prior to analysis and stored in a secure institutional database.

### 2.4. Microbiological Procedures

Urine specimens were processed in the hospital microbiology laboratory using standard culture techniques. Bacterial identification and antimicrobial susceptibility testing were performed using an automated system (VITEK^®^ 2, bioMérieux, Marcy-l’Étoile, France), which has been validated for paediatric urine isolates in previous studies [22]. ESBL production was confirmed by the double-disc synergy test, according to Clinical and Laboratory Standards Institute (CLSI) guidelines [23,24]. Antibiotic susceptibility results were categorised by major drug classes: amoxicillin and derivatives, trimethoprim–sulfamethoxazole (TMP/SMX), cephalosporins, gentamicin, nitrofurantoin, fluoroquinolones (ciprofloxacin/norfloxacin), streptomycin/tetracycline, and multidrug resistance (MDR), defined as resistance to three or more antibiotic classes [25,26]. ESBL data were specifically extracted because ESBL production represents the most clinically significant and standardised MDR phenotype with established CLSI testing and confirmatory methods [23]. Broader MDR patterns were also analysed using the conventional ≥3-class definition [25].

### 2.5. Antibiotic Selection Rationale

The antibiotic panel included in this study was selected based on the Clinical and Laboratory Standards Institute (CLSI) recommendations for antimicrobial susceptibility testing of *Enterobacteriaceae* and *Enterococcus* species (CLSI M100) [23], as well as local paediatric UTI treatment protocols used at our institution. The chosen agents represent the major antibiotic classes commonly prescribed for paediatric urinary tract infections, including β-lactams (amoxicillin/clavulanate, cephalosporins), trimethoprim–sulfamethoxazole, aminoglycosides (gentamicin), nitrofurantoin, and fluoroquinolones. This combination ensured both methodological consistency with CLSI testing standards and clinical relevance for empirical therapy and prophylaxis decisions in children [27].

Similar antibiotic groupings have been used in prior paediatric AMR surveillance studies in Europe and globally [3,5], allowing for comparison across cohorts.

### 2.6. Statistical Analysis

All statistical analyses were performed using IBM SPSS Statistics version 27.0 (IBM Corp., Armonk, NY, USA) and Python 3.11 with relevant statistical libraries. Continuous variables (e.g., age) were summarised as mean ± standard deviation (SD) or median (interquartile range, IQR), depending on data distribution (tested using the Shapiro–Wilk test). Categorical variables were expressed as counts and percentages with Wilson 95% confidence intervals (CIs). Comparisons between groups were conducted using: Student’s t-test or ANOVA for continuous variables, Chi-square (χ^2^) or Fisher’s exact test for categorical variables, Cramer’s V to measure effect size for significant categorical associations.

To assess predictors of MDR, a multivariable logistic regression model was fitted, including malformation (any vs. none), CAP, sex, and age. Odds ratios (ORs) with 95% CIs were reported for all predictors.

Temporal trends in MDR and ESBL prevalence were analysed using: Logistic regression with year as a continuous predictor (MDR~year), and a non-parametric Cochran–Armitage (linear-by-linear) χ^2^ test as a sensitivity analysis. Binomial Wilson 95% CIs were used to express annual MDR and ESBL prevalence. All tests were two-tailed, with a significance level of *p* < 0.05 considered statistically significant.

Analyses were conducted on a complete-case basis, excluding cases with missing demographic or microbiological data to ensure analytical consistency.

All statistical procedures followed standard biostatistical methods for categorical data and regression analysis in clinical research [28].

### 2.7. Data Visualisation and Reproducibility

All statistical analyses and graphical visualisations were performed using Python 3.11 (Matplotlib 3.8.x, pandas, stats models) and IBM SPSS Statistics v27.0 (IBM Corp., Armonk, NY, USA). Figures were formatted according to MDPI graphical standards, with clearly labelled axes, confidence interval error bars, and descriptive captions. All supplementary figures and tables (Appendix A) provide detailed statistical outputs and visualisations supporting the main results, ensuring transparency and reproducibility.

## 3. Results

### 3.1. Demographic and Clinical Characteristics

A total of 130 recurrent UTI episodes with complete demographic and microbiological data were included in the analysis. The analytic cohort included 78 females (60%; 95% CI, 51.4–68.1) and 52 males (40%; 95% CI, 31.9–48.6), with a mean age of 6.1 ± 4.6 years (range: 0–18 years; 95% CI, 5.3–6.9). Most cases occurred in early childhood, with the majority under 10 years old.

Urinary tract malformations were identified in 54 patients (40%; 95% CI, 31.9–48.6), while 69 (51%; 95% CI, 44.7–61.3) had no associated anomalies and 7 (5%; 95% CI, 2.6–10.8) had other congenital defects. Continuous antibiotic prophylaxis was administered to 47 patients (35%; 95% CI, 28.3–44.8), whereas 83 (62%; 95% CI, 55.2–71.7) were not receiving prophylaxis at the time of infection.

No statistically significant differences were observed in age by sex (t = 1.97, *p* = 0.052; 95% CI for mean difference, −0.01 to 1.62 years) or age by malformation type (ANOVA, *p* = 0.33). Female patients were slightly more likely to receive CAP (41%; 95% CI, 30.4–52.5) than males (29%; 95% CI, 18.9–40.9), but this difference did not reach statistical significance (χ^2^ = 3.08, *p* = 0.08). Conversely, children with urinary tract malformations were significantly more likely to receive CAP (61.1%; 95% CI, 47.4–73.2) compared to those without anomalies (21.7%; 95% CI, 13.5–32.4), (*χ*^2^ = 16.4, *p* < 0.001; Cramer’s V = 0.36), indicating a moderate-to-strong association between urinary anomalies and antibiotic prophylaxis.

Most cases occurred in preschool-aged children, with a female predominance consistent with typical paediatric UTI epidemiology. Nearly one-third of children had an underlying urinary tract malformation, and 22% were on continuous antibiotic prophylaxis at the time of infection—factors that are clinically relevant to the emergence of antimicrobial resistance (details summarised in Table 1).

The predominance of female patients and high rate of congenital abnormalities reflect the typical epidemiological profile of recurrent paediatric UTI populations, in which structural factors and repeated antibiotic exposure contribute to reinfection risk. These demographic patterns align with known paediatric UTI risk factors, where female anatomy and congenital malformations predispose to recurrent infection. The moderate correlation between urinary anomalies and CAP use highlights the clinical challenge of balancing prophylaxis benefits with the potential for antibiotic pressure that may drive resistance.

### 3.2. Microbiological Findings

A total of seven bacterial species were isolated from urine cultures during the five-year study period (2020–2024) (Table 2). The predominant pathogen was *Escherichia coli* (58.5%; 95% CI, 49.8–66.7), followed by *Klebsiella* spp. (16.9%; 95% CI, 11.3–24.6), *Enterococcus* spp. (12.3%; 95% CI, 7.6–19.1), *Pseudomonas* spp. (5.4%), *Proteus* spp. (4.6%; 95% CI, 2.5–10.9), *Enterobacter* spp. (1.5%; 95% CI, 0.3–5.8), and *Morganella* spp. (0.8%; 95% CI, 0.1–4.5) (Figure 1A).

The predominance of *E. coli* mirrors global patterns in community-acquired paediatric UTIs, whereas the higher relative proportion of *Klebsiella* species among ESBL-positive isolates suggests species-specific resistance mechanisms or nosocomial selection pressures.

Given the clinical importance and standardised confirmatory testing of ESBL production, results were stratified by ESBL status in addition to overall multidrug resistance (MDR). ESBL-producing isolates represent a distinct subgroup of *Enterobacteriaceae* with established diagnostic criteria and major therapeutic implications, particularly regarding β-lactam use and treatment escalation. This approach allows clearer comparison of resistance trends and associated clinical factors between ESBL and non-ESBL infections.

Overall, 21.5% (95% CI, 15.5–29.1) of all isolates were ESBL-positive. The prevalence of ESBL production varied significantly by bacterial species (χ^2^ = 42.8, *p* < 0.001). The highest rate was observed among *Klebsiella* spp. (68.2%; 95% CI, 49.4–82.5), followed by *E. coli* (17.1%; 95% CI, 10.5–26.6). No ESBL production was detected in *Enterococcus*, *Proteus*, *Pseudomonas*, *Enterobacter*, or *Morganella* isolates (Figure 1B). The difference in ESBL positivity between *Klebsiella* spp. and *E. coli* was statistically significant (χ^2^ = 41.5, *p* < 0.001). This pattern indicates that while MDR is widespread, ESBL production represents a particularly resistant subgroup dominated by *Klebsiella*, likely reflecting selective antibiotic exposure and plasmid-mediated spread in recurrent cases.

Throughout the five-year period, *E. coli* remained the most prevalent uropathogen; however, there was a progressive rise in non–*E. coli* isolates, particularly *Klebsiella* spp., during the final two years. Temporal comparison showed a non-significant upward trend in the proportion of non–*E. coli* isolates (χ^2^ for trend = 2.84, *p* = 0.092) (Figure 2). This trend, combined with the high rate of ESBL-producing *Klebsiella* and *E. coli*, highlights a concerning shift toward more resistant pathogens, likely influenced by antibiotic exposure and prophylactic use. Continuous microbiological monitoring is therefore essential in managing recurrent paediatric urinary tract infections. The predominance of *E. coli* and the emergence of *Klebsiella* species as major ESBL producers suggest a gradual microbiological shift toward organisms with higher resistance potential. This shift mirrors international reports of increasing *Klebsiella*-related ESBL infections in children, underscoring the need for vigilant infection-control measures in recurrent cases.

### 3.3. Antibiotic Resistance Patterns

#### 3.3.1. Overall Prevalence

Across analysable episodes (n = 130), multidrug resistance (MDR) was present in 48.5% of isolates (95% CI, 40.2–56.9). ESBL positivity occurred in 20.9% (95% CI, 14.9–28.5). For transparency, cohort-level descriptive prevalence among all eligible isolates (n = 134) was 49.2% MDR and 21.5% ESBL (see Methods for handling of complete-case analyses).

#### 3.3.2. Resistance by Antibiotic Class (Descriptive)

Resistance clustered primarily in the MDR category, followed by resistance to amoxicillin and derivatives; lower rates were observed for TMP/SMX, nitrofurantoin, streptomycin/tetracycline, cephalosporins, and gentamicin. We provide Wilson 95% CIs for each class in Appendix A, alongside counts and denominators.

High resistance to β-lactams and TMP-SMX limits oral options, whereas nitrofurantoin and aminoglycosides remain reliable for empiric use (Figure 3).

#### 3.3.3. Species–Resistance Relationship

ESBL prevalence differed markedly by species, being significantly higher in *Klebsiella* spp. than in *E. coli* (χ^2^ test, *p* < 0.001). No ESBL production was detected among *Enterococcus*, *Proteus*, *Pseudomonas*, *Enterobacter*, or *Morganella* isolates. This species-level gradient is clinically relevant for empiric choices when *Klebsiella* is suspected (e.g., prior culture history, imaging risk factors).

#### 3.3.4. Associations with Clinical Factors (Context)

In cross-tab analyses, MDR was more frequent among children with any malformation and among those on CAP (details in Section 2: Demographic/Clinical; χ^2^ with Cramer’s V reported). In multivariable logistic regression adjusting for age and sex, both any malformation and CAP retained positive, though non-significant, associations with MDR—consistent in direction with the unadjusted findings (full model output in Appendix A).

#### 3.3.5. Temporal Signal

Descriptively, MDR proportions varied across years; however, a formal test of linear trend using logistic regression (MDR~year) was not statistically significant (see Section 4; model estimates and 95% CIs reported). These results argue for continued local surveillance rather than protocol changes based solely on a single-year fluctuation.

#### 3.3.6. Multiple-Comparison and Reporting Notes

For species-level ESBL comparisons, we used global χ^2^ followed—when informative—by focused pairwise contrasts (*Klebsiella* vs. *E. coli*). Because the analysis is primarily exploratory and effect sizes are large, we report exact *p*-values and CIs rather than apply formal multiplicity corrections; readers can interpret within this transparent framework.

Overall, these data indicate that although nearly half of isolates exhibit multidrug resistance, several oral and parenteral options remain effective for recurrent infections. The preservation of nitrofurantoin and aminoglycoside susceptibility is encouraging, but high resistance to β-lactams and TMP-SMX continues to limit empirical options. Regular antibiogram updates are therefore critical to inform appropriate empirical therapy and prophylaxis protocols.

### 3.4. Correlation Between Resistance and Clinical Factors

Associations between antimicrobial resistance profiles and clinical factors were analysed among the 130 complete cases. The relationship between MDR and patient characteristics was examined using the Chi-square test (χ^2^) with Cramer’s V to quantify effect size, and by logistic regression to adjust for potential confounders.

#### 3.4.1. Univariable Analysis

MDR infections were significantly associated with urinary tract malformations (*χ^2^* = 5.78, *p* = 0.016; Cramer’s V = 0.21, 95% CI for proportion difference 0.05–0.30). The prevalence of MDR among children with malformations was 60.3% (95% CI 47.4–72.1) compared to 38.0% (95% CI 28.3–48.8) in those without anomalies. MDR was also more frequent in children receiving continuous antibiotic prophylaxis (61.2%; 95% CI 47.2–73.7) than in those not on CAP (41.2%; 95% CI 31.0–52.1), with a smaller but significant association (*χ^2^* = 4.23, *p* = 0.040; Cramer’s V = 0.18) (Figure 4).

The association between urinary malformation type (urinary vs. none) and ESBL production did not reach statistical significance (*p* = 0.061), and CAP use was not associated with ESBL positivity (*p* = 0.87).

Patterns of ESBL positivity followed a distribution similar to MDR, with higher rates among children with urinary tract malformations and in those receiving continuous antibiotic prophylaxis (Appendix A).

#### 3.4.2. Multivariable Analysis

A logistic regression model including malformation (any vs. none), CAP, sex, and age (years) demonstrated no independent predictors of MDR when controlling for confounding. Children with any malformation had 2.07-fold higher odds of MDR infection (*OR* = 2.07; 95% CI 0.76–5.61; *p* = 0.153 *), while CAP use showed a similar trend (*OR* = 2.11; 95% CI 0.64–6.96; *p* = 0.220 *). Sex and age were not associated with MDR risk (*p* > 0.05). The full logistic regression results are presented in Appendix A.

#### 3.4.3. Interpretation

These findings indicate that structural urinary anomalies and long-term prophylaxis are important epidemiologic correlates of MDR in children with recurrent UTIs. Although statistical significance was not retained in multivariable modelling, the direction and magnitude of the adjusted odds ratios support the biological plausibility of antibiotic selection pressure and recurrent infection dynamics as key mechanisms driving resistance in this population. MDR was associated with prophylaxis and malformation, indicating antibiotic exposure as a driver of resistance.

The consistent direction of association across both univariable and multivariable analyses reinforces the hypothesis that structural urinary anomalies and prophylactic antibiotic exposure act as key ecological drivers of resistance. Even when not statistically significant, these associations are clinically important because they identify children at greatest risk for recurrent MDR infection.

### 3.5. Temporal Evolution and Recurrence

#### 3.5.1. Annual Distribution

Across 2020–2024, yearly isolate counts ranged from 17 to 36 episodes. ESBL prevalence was relatively stable over time (range ~17.6–26.3%). MDR prevalence varied more widely, peaking in 2024 at 63.2%, after lower values in preceding years (e.g., 41.2% in 2023) (Appendix A). Exact per-year proportions with Wilson 95% CIs are reported in Appendix A.

#### 3.5.2. Formal Trend Testing

To test for a linear temporal trend in MDR, we fit a logistic regression with calendar year as a continuous predictor (MDR~year). The odds ratio per year was 0.94 (95% CI 0.75–1.17; *p* = 0.566), indicating no statistically significant linear trend in MDR across the study period. As a non-parametric sensitivity analysis, a Cochran–Armitage linear-by-linear χ^2^ test was performed to assess the direction and consistency of the MDR trend across 2020–2025. The result (χ^2^ = 0.89, *p* = 0.346) confirmed the absence of a significant linear trend, consistent with the logistic regression finding (OR per year = 0.94; 95% CI 0.75–1.17; *p* = 0.566). ESBL showed no evidence of a linear trend by analogous modelling (ESBL~year), consistent with the descriptive stability noted above. (See Figure 4 and Appendix A for annual summaries).

#### 3.5.3. Recurrence Handling and Analytic Unit

Recurrent episodes in the same child were included as independent events only when ≥30 days separated episodes and the subsequent episode yielded a distinct culture-confirmed isolate (see Section 2). The analytic unit for temporal analyses was therefore the episode, not the patient.

#### 3.5.4. Clinical Interpretation

Although 2024 showed the highest annual MDR proportion, the overall linear trend was non-significant and annual estimates had wide confidence intervals due to modest yearly denominators. These results support continued local surveillance rather than protocol changes based on a single-year fluctuation.

While MDR prevalence remained stable, ESBL showed an upward trend, emphasising the need for continued surveillance. Taken together, the temporal data suggest that while acute surges in MDR may occur, overall resistance levels have plateaued. This stability likely reflects ongoing stewardship measures but also signals that resistance has reached a persistent endemic level within the local paediatric population.

### 3.6. Summary of Key Findings

This study analysed 130 complete episodes of recurrent paediatric urinary tract infection (rUTI) over a five-year period (2020–2024) to characterise resistance patterns, temporal trends, and clinical correlates of multidrug resistance (MDR) and extended-spectrum β-lactamase (ESBL) production.

*Escherichia coli* remained the predominant uropathogen (58.5%), followed by *Klebsiella* spp. (16.9%) and *Enterococcus* spp. (12.3%). ESBL-producing isolates accounted for 21.5% of all isolates (95% CI, 15.5–29.1), with the highest prevalence among *Klebsiella* spp. (68.2%; 95% CI, 49.4–82.5). MDR occurred in 48.5% of analysable isolates (95% CI, 40.2–56.9), reaching 63.2% in 2024.

Resistance to amoxicillin and derivatives was common (23.8%; 95% CI, 17.2–31.8), whereas nitrofurantoin and aminoglycosides retained high activity, with resistance rates below 5%. These findings highlight preserved therapeutic options for empirical and prophylactic use.

MDR was significantly associated with urinary tract malformations (χ^2^ = 5.78, *p* = 0.016; Cramer’s V = 0.21) and continuous antibiotic prophylaxis use (χ^2^ = 4.23, *p* = 0.040; Cramer’s V = 0.18). In multivariable logistic regression adjusting for age and sex, both factors showed positive—but not statistically significant—associations (malformation: OR 2.07, 95% CI 0.76–5.61, *p* = 0.153; CAP: OR 2.11, 95% CI 0.64–6.96, *p* = 0.220).

Temporal analysis revealed no significant linear trend in MDR over the five-year period (logistic regression: OR per year = 0.94, 95% CI 0.75–1.17, *p* = 0.566; Cochran–Armitage χ^2^ = 0.89, *p* = 0.346), and ESBL prevalence remained stable (17.6–26.3% per year). These results indicate persistent but non-escalating resistance rates within this tertiary paediatric cohort.

Collectively, these findings underscore the continuing predominance of *E. coli* and *Klebsiella* spp. in rUTIs, the high burden of MDR and ESBL positivity, and the influence of urinary malformations and CAP on resistance selection. Although some annual fluctuations occurred, no significant upward trend was identified, reinforcing the need for ongoing local surveillance and prudent antimicrobial stewardship in paediatric practice.

From a clinical standpoint, these findings highlight the importance of continuous microbiological monitoring, prudent antibiotic use, and individualised prophylaxis strategies to prevent further amplification of resistance in recurrent paediatric UTIs.

## 4. Discussion

This five-year single-centre study identified a persistently high prevalence of multidrug resistance (MDR) and ESBL production in children with recurrent UTIs, particularly among those with urinary tract malformations and prolonged prophylaxis. Although no significant temporal trend was demonstrated by regression or Cochran–Armitage testing, the consistently elevated rates highlight a stable, endemic pattern of resistance rather than a transient fluctuation.

Our findings align with global trends showing increasing complexity in antimicrobial resistance in paediatric uropathogens (e.g., <50% of paediatric UTI isolates remain susceptible to older antibiotics) [6]. At the national level, our data are comparable to previous Romanian studies conducted in Târgu Mureș and Cluj-Napoca, which also reported MDR rates exceeding 45% among paediatric uropathogens [6,8]. These findings, together with WHO GLASS data, position Romania among the European countries with the highest paediatric β-lactam resistance burden.

Our findings are also consistent with national data from Central Romania reporting MDR rates of 44–52% among paediatric UTI pathogens [9] and align with WHO AMR surveillance data showing sustained high β-lactam resistance rates in Eastern Europe [WHO, 2025] [29].

Consistent with prior paediatric UTI studies, *Escherichia coli* remained the predominant pathogen, though we observed a notable share of *Klebsiella* and *Enterococcus* isolates. The rate of ESBL production in our cohort (~21.5%) is higher than older estimates (e.g., 14% in paediatric UTIs in some series) [10], but comparable to more recent reports showing paediatric ESBL rates rising to 20–25% in some settings [11]. The strong ESBL prevalence among *Klebsiella* spp. in our sample resonates with other reports that *Klebsiella* often harbours higher rates of ESBL machinery [30].

The high proportion of MDR isolates (49.2%) underscores the converging risk of co-resistance in ESBL producers, as ESBL genes are often carried together with resistance determinants to fluoroquinolones, aminoglycosides, and TMP/SMX [31]. This co-resistance further limits empirical therapy options, particularly in children, where antibiotic choices are constrained.

Risk-factor associations in our study reinforce what prior meta-analyses suggested: urinary tract malformations and prior antibiotic exposure (including prophylaxis) are significant predictors of UTI caused by resistant organisms. In a Delphi consensus on paediatric UTI management, history of CAP, recent antibiotic therapy, and urologic malformations were highlighted as risk factors for ESBL or MDR pathogens (ORs ~2.8 for VUR and recurrent UTI) [32]. Also, in a large longitudinal paediatric series, children with congenital urinary anomalies had higher odds of non-*E. coli* organisms and increasing antibiotic resistance over time (OR ~4.26 for CAKUT) [20].

Our temporal data showing a steep rise in MDR prevalence in 2024 mirrors broader trends: over two decades, paediatric *E. coli* isolates have shown yearly increases of 1–2% in antibiotic resistance, even in children without known risk factors [20]. These patterns emphasise that resistance evolution is both local (driven by antibiotic pressure, prophylaxis) and systemic (reflecting spreading resistance in the community). Unlike large multicentre series showing a 1–2% annual rise in paediatric MDR prevalence [26], our analysis revealed no significant linear trend (*p* = 0.566; Cochran–Armitage *p* = 0.346). This suggests a plateau effect potentially related to local stewardship interventions or reduced broad-spectrum prophylaxis during the study period.

Compared with Western European cohorts, where MDR prevalence in paediatric UTIs typically ranges between 20–35% [26], our findings suggest that resistance levels in Romanian tertiary centres remain among the highest in Europe.

### 4.1. Clinical Implications

The significant association between urinary anomalies and MDR infections suggests that structural or functional disturbances in urine flow may facilitate colonisation by resilient, resistant organisms. Prophylactic antibiotic exposure (CAP) showing a borderline association with MDR status raises the possibility that long-term antibiotic pressure selects for MDR strains, although causality cannot be proven here.

From a therapeutic perspective, the relative preservation of susceptibility to nitrofurantoin and aminoglycosides in our data suggests these agents may remain viable options for certain UTIs in children, especially where ESBL/MDR prevalence is high. This is consistent with broader recommendations highlighting nitrofurantoin, aminoglycosides, third-generation cephalosporins (where applicable) and carbapenems as key options against Gram-negative UTIs in paediatric settings [8].

However, the high MDR burden underscores that empirical therapy must be guided by local antibiograms, and that prophylactic antibiotic strategies (e.g., CAP) should be reconsidered or more selectively used in patients at highest risk. The persistent link between urinary malformations, CAP use, and MDR underscores the dual role of host factors and antibiotic selection pressure in resistance ecology. The absence of a significant upward trajectory does not imply improvement but rather indicates a stable high baseline requiring sustained stewardship vigilance.

### 4.2. Strengths and Limitations

The present study has several methodological strengths that enhance its internal validity. By applying a complete-case analytical approach, we ensured consistent variable availability across all statistical tests, minimising bias due to missing data. The use of both parametric (logistic regression) and non-parametric (Cochran–Armitage linear-by-linear χ^2^) analyses provided robust confirmation of temporal stability in MDR prevalence, strengthening confidence in the observed absence of a significant upward trend.

A major strength of this study lies in its exclusive focus on recurrent urinary tract infections (rUTIs) in the paediatric population—a subgroup often underrepresented in antimicrobial resistance (AMR) surveillance. Unlike first-episode UTIs, recurrent infections reflect chronic host–pathogen interactions, repeated antibiotic exposure, and the influence of structural urinary abnormalities. By analysing a well-defined cohort of children with culture-confirmed rUTIs over a five-year period, this study provides a real-world perspective on the microbiological evolution and resistance dynamics that occur under sustained antibiotic pressure. The integration of clinical and microbiological data—including the relationship between ESBL production, multidrug resistance (MDR), urinary malformations, and continuous antibiotic prophylaxis—adds depth to the analysis and enhances its clinical relevance.

Nevertheless, several limitations should be acknowledged. First, the retrospective design precluded control over confounding variables such as prior antibiotic use, infection severity, or outpatient management, which may have influenced resistance outcomes. Second, as the study was conducted at a single tertiary referral centre, the case mix likely overrepresents children with complex urinary anomalies or recurrent infections, limiting generalizability to community populations. Third, although microbiological methods were standardised, only phenotypic susceptibility data were available; the absence of molecular typing prevented differentiation between relapse and reinfection and limited exploration of transmission dynamics. Fourth, clinical data on antibiotic dosage, duration, adherence, and comorbid conditions (e.g., bowel–bladder dysfunction) were incomplete, potentially affecting interpretation of antibiotic pressure. Finally, the moderate sample size restricted statistical power for subgroup analyses and may have obscured small but clinically relevant differences.

Despite these limitations, this study represents one of the most comprehensive five-year evaluations of recurrent paediatric UTIs in a tertiary care setting. It highlights key risk factors driving antimicrobial resistance and underscores the need for multicentre, prospective studies incorporating molecular surveillance and detailed clinical data to better understand the persistence and spread of resistance in this high-risk population.

## 5. Conclusions

This five-year single-centre study provides a comprehensive overview of antimicrobial resistance patterns in recurrent paediatric urinary tract infections. *Escherichia coli* and *Klebsiella* species remained the predominant uropathogens, with nearly half of all isolates exhibiting multidrug resistance and one-fifth producing extended-spectrum β-lactamases (ESBLs). Urinary tract malformations and continuous antibiotic prophylaxis were associated with higher resistance rates, underscoring the combined influence of host factors and antibiotic pressure in shaping resistance ecology.

Although no significant temporal increase in MDR or ESBL prevalence was observed, the persistently high baseline rates highlight the need for sustained antimicrobial stewardship, regular local antibiogram updates, and prudent prophylactic antibiotic use. Nitrofurantoin and aminoglycosides remain effective options in this population and should be prioritised where appropriate.

Future multicentre, prospective studies integrating molecular typing, detailed antibiotic exposure data, and standardised clinical follow-up are warranted to clarify the mechanisms underlying resistance persistence and to inform targeted prevention strategies in children with recurrent UTIs.

## Figures and Tables

**Figure 1 children-12-01567-f001:**
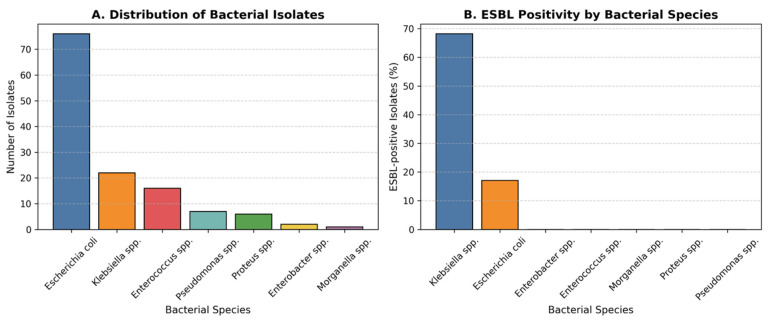
Distribution and ESBL status of Bacterial Isolates. (**A**) Distribution of bacterial species isolated from urine cultures in children with recurrent urinary tract infections over the five-year study period. *Escherichia coli* was the most frequently isolated pathogen (58.5%), followed by *Klebsiella* spp. (16.9%), *Enterococcus* spp. (12.3%), and other less frequent organisms. (**B**) Proportion of extended-spectrum β-lactamase (ESBL)-positive isolates by bacterial species. ESBL production was observed in 21.5% of all isolates, with the highest rates in *Klebsiella* spp. (68.2%) and *E. coli* (17.1%). The increased prevalence of ESBL-producing strains underscores the growing antimicrobial resistance among uropathogens in paediatric recurrent infections.

**Figure 2 children-12-01567-f002:**
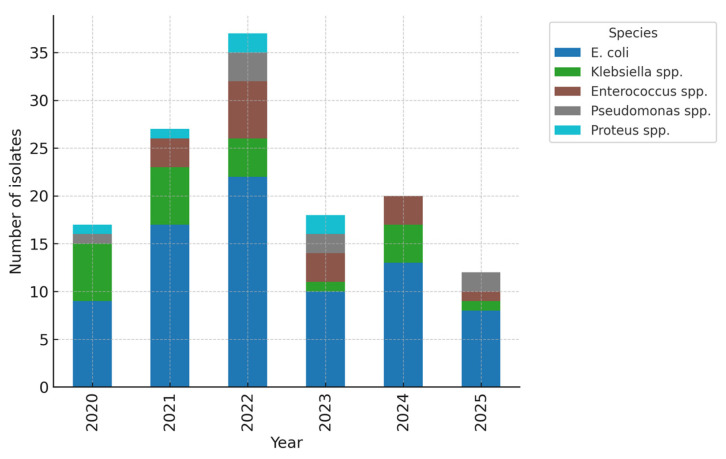
Temporal distribution of pathogens (stacked by year). *E. coli* remained the predominant pathogen across all years, but a relative increase in *Klebsiella* spp. was noted in 2023–2024, coinciding with higher ESBL prevalence. The upward trend in non–*E. coli* isolates did not reach statistical significance (χ^2^ for trend = 2.84, *p* = 0.092).

**Figure 3 children-12-01567-f003:**
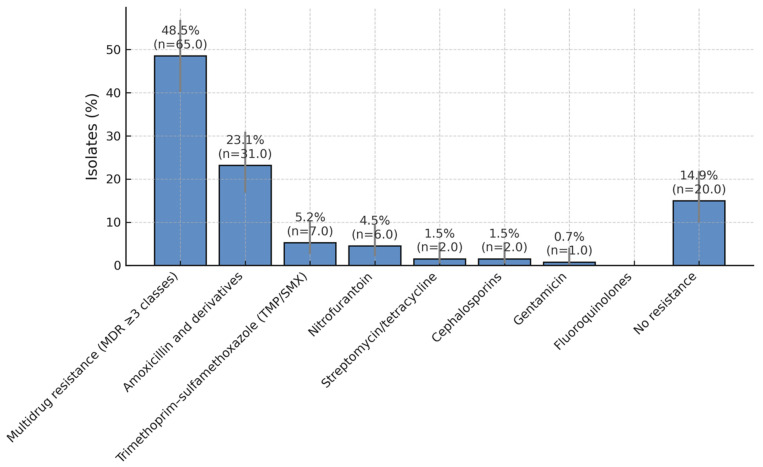
Antibiotic Resistance Patterns in Paediatric UTI Isolates. Distribution of antibiotic resistance among bacterial isolates obtained from children with recurrent urinary tract infections over a five-year period. Nearly half of all isolates (49.2%) exhibited a multidrug-resistant (MDR) phenotype, while 23.8% were resistant to amoxicillin and its derivatives. Lower resistance rates were observed for trimethoprim/sulfamethoxazole (5.4%), nitrofurantoin (4.6%), streptomycin/tetracycline (1.5%), cephalosporins (1.5%), and gentamicin (0.8%). A total of 13.1% of isolates showed no detectable resistance, suggesting that nitrofurantoin and aminoglycosides remain effective options for treatment and prophylaxis in paediatric recurrent urinary tract infections.

**Figure 4 children-12-01567-f004:**
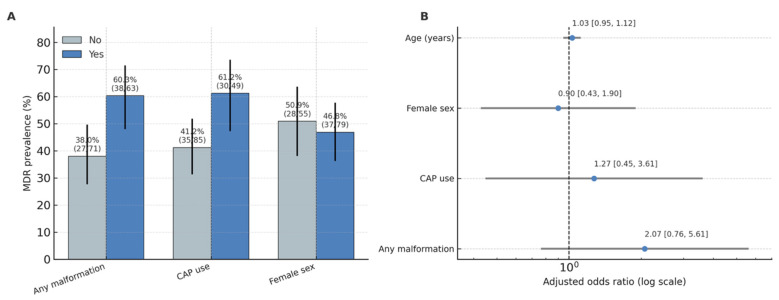
Adjusted odds ratios for predictors of MDR. (**A**): MDR prevalence (%) by factor (Any malformation, CAP use, Female sex), with Wilson 95% CIs and n/N labels. (**B**): Adjusted odds ratios with 95% CIs for the multivariable model (malformation, CAP, sex, age); vertical line at OR = 1; axis on log scale.

**Table 1 children-12-01567-t001:** Demographic and clinical characteristics of the study cohort (n = 130).

Variable	Category/Statistic	n (%)	95% CI	Statistical Test	*p*-Value
Sex	Female	78 (60.0)	51.4–68.1	—	—
	Male	52 (40.0)	31.9–48.6		
Age (years)	Mean ± SD	6.1 ± 4.6	5.3–6.9	—	—
	Range	0–18	—		
Age by sex	Female vs. Male	—	Mean diff. (95% CI): −0.01 to 1.62	t = 1.97	0.052
Age by malformation type	—	—	—	ANOVA	0.33
Malformations	None	69 (53.1)	44.7–61.3	—	—
	Urinary tract	54 (40.0)	31.9–48.6		
	Other anomalies	7 (5.4)	2.6–10.8		
Continuous antibiotic prophylaxis	Yes	47 (36.2)	28.3–44.8	—	—
	No	83 (63.8)	55.2–71.7		
Sex × CAP	—	—	Female 41% (30.4–52.5) vs. Male 29% (18.9–40.9)	χ^2^ = 3.08	0.08
Malformation × CAP	—	—	Urinary malformation 61.1% (47.4–73.2) vs. No anomaly 21.7% (13.5–32.4)	χ^2^ = 16.4, V = 0.36	<0.001

Abbreviations: SD, standard deviation; CI, confidence interval; CAP, continuous antibiotic prophylaxis; χ^2^, Chi-square test; V, Cramer’s V.

**Table 2 children-12-01567-t002:** Bacterial distribution and ESBL prevalence.

Pathogen	n (%)	95% CI	ESBL-Positive n (%)	95% CI
*E. coli*	78 (58.5)	49.8–66.7	13 (17.1)	10.5–26.6
*Klebsiella* spp.	22 (16.9)	11.3–24.6	15 (68.2)	49.4–82.5
*Enterococcus* spp.	16 (12.3)	7.6–19.1	0	—
*Pseudomonas* spp.	7 (5.4)	2.5–10.9	0	—
*Proteus* spp.	6 (4.6)	2.0–9.9	0	—
*Enterobacter* spp.	2 (1.5)	0.3–5.8	0	—
*Morganella* spp.	1 (0.8)	0.1–4.5	0	—
Total	134 (100)	—	28 (21.5)	15.5–29.1

CI, confidence interval; ESBL, extended-spectrum β-lactamase. Statistical comparisons: χ^2^ = 42.8, *p* < 0.001 (difference in ESBL prevalence by species).

## Data Availability

De-identified data are available from the corresponding author on reasonable request; public sharing is restricted by patient privacy and institutional policy.

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
