# Peer review of "Antibiotic Resistance Trends in Recurrent Paediatric Urinary Tract Infections: A Five-Year Single-Centre Experience"

_children, 2025, doi:10.3390/children12111567_

Round 1
Reviewer 1 Report
Comments and Suggestions for Authors
Introduction:
1. Please add more information regarding ESBL in pediatrics.
Method:
1. Please add reference to the inclusion/exclusion
2. What is the explanation of incomplete microbiology data?
3. What is the definition of contaminated samples?
4. What is the reasons for extracted data only for ESBL? Others MDR pattern?
5. What is the reasons for choosing reported antibiotic in this study? Antibiotic is based on CLSI or clinical guideline? Please explain more clearer.
6. Line 139: Need references for this sentences.
7. Line 140-143: Need references.
8. Method section is lack of references.
Result:
1. Line 191-193 is replicated with line 109-111, about the initial screening number. Author need to choose to put the information, in the method section or result section.
2. In the result, what is the reasons stratification for ESBL? Not MDR at general ?
3. Result section need to be more comprehensive written. It is good to have separate subheadings, however there is lack of comprehensive explanation in the result section.
Discussion:
1. What is the section of comparison with literature? This subheading is confusing.
2. Line 495: this study is observing the recurrent UTIs, therefore severe case or complicated ones will occurs and this seem not adequate for limitation reasoning. Author need to observe more about the limitation of the study.
Reviewer 2 Report
Comments and Suggestions for Authors
I have reviewed the manuscript entitled "Antibiotic Resistance Trends in Recurrent Paediatric Urinary Tract Infections: A Five-Year Single-Centre Experience" written by Olivia Stanciu, Mircea Andriescu, Andreea Moga, Ruxandra Caragata and Radu Balanescu. The topic is a very intriguing one and the present study manages to discuss various aspects on the topic.
The authors hypothesize that children with urinary tract malformations and those receiving antibiotic prophylaxis would exhibit higher rates of multidrug-resistant and extended-spectrum beta-lactamase producing uropathogens. Also the authors emphasize that understanding local antibiotic resistance trends represents a key focus in children.
I recommend the authors to use mor significant references on the same topic as there are antibiotic resistance studies in Romania focusing on CAKUT patients only.
Introduction section is comprehensive and mainly describes the differences between first episode of UTI and recurrent UTIs., the difficulty in managing recurrent UTIs and present national knowledge in the field. There are Romanian studies focusing on antibiotic resistance patterns of uropathogens causing UTI in children with CAKUT. What are the main causes for recurrent UTIs in children? What types of CAKUT are associated with recurrent UTI? Is it possible to add the specific data in the Introduction section? What are other causes of recurrent UTI beside presence of specific CAKUT types?
Are there 130/134 patients? or urinary tract infections episodes? Please state the difference between the two entities.
The materials and methods section describes how the study was designed, the data collection method, microbiological procedures. There is a small missunderstanding in the number of the eligible patients. What is the correct number of patients within the study? 130? 134? The results section should be adapted accordingly to this number.
How was presence and type of CAKUT established? Was abdominal ultrasound performed in all patients? Was the diagnosis of CAKUT already known? What type of CAKUT was of interest to the researchers?
The abbreviations need to be explained only at their first text appearance in the text. See CAP for example.
In the materials and methods section there is no need to make a brief preview of the analytical components of the study (see lines 168-184).
I would recommend renaming the Figure 1 in accordance with its graphics. There is no evidence on the two figures of resistance profile of bacterial isolates.
Line 264 first mentions MDR as present in 48.5% of isolates. I suggest adding some details in the Introduction section regarding MDR definition and characteristics.
I suggest the authors to decide upon the correct number of participants in the study (130 or 134? - line 260) and use only one number, in order not to create confusion.
Line 271: there is no need for "To meet MDPI reporting standards"
The Discussion section needs improvements. The authors need to compare own results with already published literature results starting with the close neighborhood of study location, same city, same country (romanian studies), followed by generalization to Europe and worldwide. (WHO statistics).
The conclusion are supported by the results.
In the beginning of the manuscript I suggest starting with the Abstract. The highlights of the study are not necessarily needed in the beginning of the document.
Congratulations to the authors for entire work.
Author Response
Please see the attachament.

Round 2
Reviewer 1 Report
Comments and Suggestions for Authors
Author has addressed the previous comment properly, no outstanding question regarding the second round review. Please check any typing/spelling error that still exist, since author used AI to refine the language/structure of the manuscript.